# Cooperation in alternating interactions with memory constraints

Peter S. Park [1], Martin A. Nowak [1,2] & Christian Hilbe [3✉]

In repeated social interactions, individuals often employ reciprocal strategies to maintain cooperation. To explore the emergence of reciprocity, many theoretical models assume synchronized decision making. In each round, individuals decide simultaneously whether to cooperate or not. Yet many manifestations of reciprocity in nature are asynchronous. Individuals provide help at one time and receive help at another. Here, we explore such alternating games in which players take turns. We mathematically characterize all Nash equilibria among memory-one strategies. Moreover, we use evolutionary simulations to explore various model extensions, exploring the effect of discounted games, irregular alternation patterns, and higher memory. In all cases, we observe that mutual cooperation still evolves for a wide range of parameter values. However, compared to simultaneous games, alternating games require different strategies to maintain cooperation in noisy environments. Moreover, none of the respective strategies are evolutionarily stable.

[1] Department of Mathematics, Harvard University, Cambridge, MA 02138, USA. [2] Department of Organismic and Evolutionary Biology, Harvard University, Cambridge, MA 02138, USA. [3] Max Planck Research Group Dynamics of Social Behavior, Max Planck Institute for Evolutionary Biology, 24306 Plön, Germany. ✉email: hilbe@evolbio.mpg.de

Cooperation can be maintained by direct reciprocity, where individuals help others in repeated interactions[1–3]. Traditionally, researchers capture the logic of direct reciprocity with the repeated prisoner's dilemma[4–17]. According to that model, two individuals—usually referred to as players—interact with each other over several rounds. In each round, both players can either cooperate or defect. Mutual cooperation yields a better payoff than mutual defection, but each individual has an incentive to defect. Theoretical and experimental work suggests that cooperation can evolve if there are sufficiently many interactions between the individuals[18]. This work has been used to explain a wide variety of behaviors, including why humans are more likely to cooperate in stable groups[19], why certain animal species share food[20], and why firms are able to achieve higher market prices when they engage in collusion[21].

A standard assumption that underlies much of this research is that individuals make their decisions simultaneously (or at least in ignorance of the co-player's current decision). We refer to this kind of repeated interaction as a simultaneous game (Fig. 1a). For many natural manifestations of reciprocity, however, simultaneous cooperative exchanges are unlikely or even impossible, such as when people ask for favors[22], vampire bats donate blood to their conspecifics[20], sticklebacks engage in predator inspection[23], or ibis take turns when leading their flock[24]. Such interactions are better captured by alternating games, in which players consecutively decide whether to cooperate[25–28]. When individuals decide asynchronously, they make their decisions based on different histories. The most recent events one player has in memory differ from the most recent events that the next player takes into account (Fig. 1b). Such asymmetries in turn make it more difficult to successfully coordinate on cooperation. As a result, many well-known strategies like Tit-for-Tat or Win-Stay Lose-Shift fail to evolve when players move alternatingly[25,26]. Instead, previous computational[25–27] and experimental studies[29] suggest that individuals need to be more forgiving. However, a full understanding of optimal play in alternating games is lacking, even though optimal behavior in the simultaneous game is by now well-analyzed[30–38].

Here, we propose an analytical approach to describe when cooperation evolves in the alternating game. In line with the previous literature, we typically focus on individuals with so-called memory-one strategies[3]. Memory-one strategies depend on each player's most recent move. Our analysis involves two steps. First, we show that successful play in alternating games does not require a sophisticated cognitive apparatus. More specifically, when interacting with a given memory-one opponent, it suffices to respond with a reactive strategy that only depends on the co-player's most recent move. This result is reminiscent of a previous finding of Press and Dyson for the simultaneous game[39]. They showed that against a memory-one strategy, there is nothing to gain from having a larger memory than the opponent. Our result for the alternating game goes one step further. Against a memory-one strategy, players can afford to have a strictly lower memory, without any loss to their or their co-player's payoff. As we show, this result crucially depends on the alternating move structure; it is not true when players move simultaneously. In a second step, we show that in order to identify the best response to a given memory-one player, we only need to check the four most extreme reactive strategies: unconditional defection, unconditional cooperation, Tit-for-Tat, and Anti-Tit-for-Tat. Using this approach, we identify all Nash equilibria among the memory-one strategies.

In the absence of errors, we find an unexpected equivalence. The very same memory-1 strategies that can be used to enforce cooperation in the simultaneous game also enforce cooperation in the alternating game. However, once we take into account errors, the predictions for the two models diverge. In the simultaneous game, Win-Stay Lose-Shift is evolutionarily stable when the benefit to cost ratio is sufficiently large and when errors are sufficiently rare[40,41]. In that case, there is a simple rule for how to sustain full cooperation: individuals should repeat their previous action if it yielded a sufficiently large payoff, and switch to the opposite action otherwise. In contrast, in the alternating game, all stable cooperative strategies require players to randomize. After mutual defection, they need to cooperate with some well-defined probability that depends on the game parameters and the error rate. Although the respective strategies in the alternating game are Nash equilibria, we show that none of them is evolutionarily stable. As a result, evolving cooperation rates in the alternating game often tend to be lower than in the simultaneous game, although this difference is smaller than perhaps expected from static stability considerations alone. We summarize our analytical findings in Fig. 2.

Our work suggests that in most realistic scenarios, successful play in alternating games requires different kinds of behaviors than predicted by the earlier theory on simultaneous games. In this way, we corroborate earlier experimental work on human cooperation[29] and provide theoretical methods to further analyze repeated games in the future. Overall, we find that cooperation is still feasible in alternating games. However, the strategies that enforce cooperation

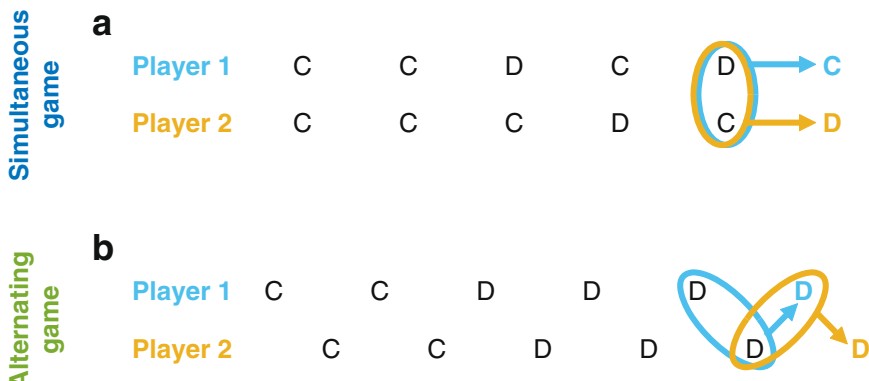

**Fig. 1 Game dynamics for the simultaneous and the alternating game.** In both the simultaneous and the alternating game, two players interact repeatedly. In each turn, they decide whether to cooperate (*C*) or to defect (*D*). In the simultaneous game (**a**), they make their decision at the same time (or at least not knowing the other player's decision). In the alternating game (**b**), one player decides before the other player does. In both cases, we study memory-1 strategies. That is, an individual's next action only depends on each individual's previous action. We illustrate the information each individual takes into account for their last decision with colored ellipses. In the simultaneous game, individuals take into account the same information. In the alternating game, decisions are based on different sets of information.

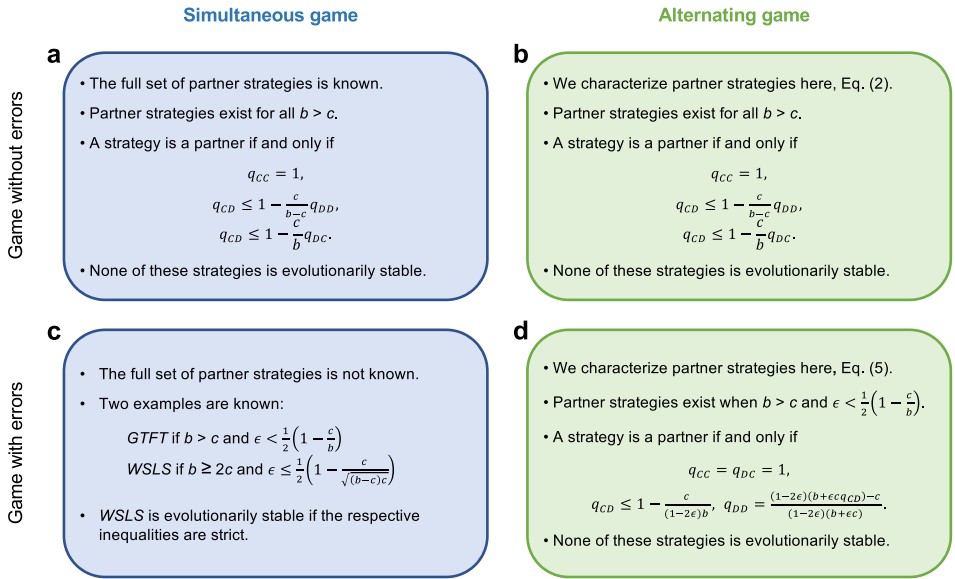

**Fig. 2 A characterization of partners among the memory-1 strategies.** Within the class of memory-1 strategies, we provide an overview of the strategies that sustain full cooperation in a Nash equilibrium. The respective strategies are called partner strategies, or partners[18]. **a** For the simultaneous game without errors, partners have been first described by Akin[34,35] (he calls them "good strategies"). Akin's approach has been extended by Stewart and Plotkin[31] to describe all memory-1 Nash equilibria of the simultaneous game. In the absence of errors, none of these strategies is evolutionarily stable[55,57]. Instead, one can always find neutral mutant strategies which act as a stepping stone out of equilibrium[58]. **b** For the alternating game without errors, Eq. (2) provides a full characterization of all partner strategies. Cooperation is maintained by the same strategies as in the simultaneous game. **c** Despite decades of research, the exact set of partner strategies for the simultaneous game with errors is not known. However, there are at least two instances of partner strategies, GTFT[6,49], and Win-Stay Lose-Shift, WSLS[42,53]. For repeated games with errors, evolutionary stability is generally feasible[56]. In particular, WSLS is evolutionarily stable if the benefit to cost ratio is sufficiently large and if errors are sufficiently rare[40]. **d** For the alternating game with errors, we characterize all partner strategies in Eq. (5). None of them is deterministic. As a result, none of them is evolutionarily stable (see Supplementary Information for details).

can be neutrally invaded, and hence cooperation tends to be more short-lived than in the simultaneous game.

## Results

**Model description**. In the following, we formulate a simple baseline scenario, which we use to derive our main analytical results (see also Supplementary Note 1). More general scenarios are discussed in a later section, and in full detail in Supplementary Note 3. We consider interactions between two players, player 1 and player 2. Both players repeatedly decide whether to cooperate ($C$) or defect ($D$). These repeated interactions can take place in two different ways. In the *simultaneous game*, there is a discrete number of rounds. In each round, both players make their decision at the same time, not knowing their co-player's decision (Fig. 1a). In contrast, in the alternating game, the players move consecutively. We consider the strictly alternating game: Player 1 moves first, and then player 2 learns about player 1's decision and moves next (Fig. 1b). We note that there are also variants of the alternating game in which the order of moves is random[25,28]. In particular, one player may by chance make two or more consecutive moves before it is the other player's turn again. The effect of such irregular alternation patterns will be discussed later.

For the simultaneous game, the possible payoffs in each round can be represented by four parameters. Players receive the reward $R$ in rounds in which they both cooperate; they receive the temptation payoff $T$ and the sucker's payoff $S$, respectively, if only one player cooperates; and they receive the punishment payoff $P$ in case they both defect. For $T > R > P > S$, we obtain the prisoner's dilemma. In the alternating game, however, it is useful to assume that payoffs can be assigned to each player's individual action[25]. In that case, the value of one player's cooperation is independent of the co-player's previous or subsequent decision

(or equivalently, payoffs are independent of how the two players' decisions are grouped into rounds). As a result, we obtain the donation game[3]. Here, cooperation means to pay a cost $c > 0$ in order to provide a benefit $b > c$ to the co-player. The donation game is a special case of a prisoner's dilemma for which

$$R = b - c, \quad S = -c, \quad T = b, \quad P = 0. \qquad (1)$$

To compare the alternating game with the simultaneous game, we assume payoffs satisfy (1) throughout.

In the baseline scenario, we consider infinitely repeated games, and we study players who make their decisions based on each player's most recent move. In the simultaneous game, the respective strategies are called memory-1 strategies[42]; they take into account the outcome of one previous round (Fig. 1a). Such strategies can be represented as a 4-tuple, $\mathbf{p} = (p_{CC}, p_{CD}, p_{DC}, p_{DD})$. The entry $p_{ij}$ denotes the probability to cooperate in the next round. This probability depends on the player's action $i$ and the co-player's action $j$ in the previous round. The equivalent strategy class also exists in alternating games[25]. In alternating games, however, there is no longer a unique previous round to which both players refer. Instead, the last round that is taken into account depends on the perspective of each player. It consists of the respective last moves of the two players (Fig. 1b). An important subset of memory-1 strategies is the set of so-called reactive strategies. Here, players ignore their own previous action and only condition their behavior on what the co-player previously did. Reactive strategies are therefore those memory-1 strategies for which $p_{CC} = p_{DC}$ and $p_{CD} = p_{DD}$.

Some well-known examples of memory-1 strategies for the simultaneous game include Always Defect, $ALLD = (0, 0, 0, 0)$, Tit-for-Tat, $TFT = (1, 0, 1, 0)$, and Win-Stay Lose-Shift, $WSLS = (1, 0, 0, 1)$. In the alternating game, a strategy called Firm-but-Fair[3],

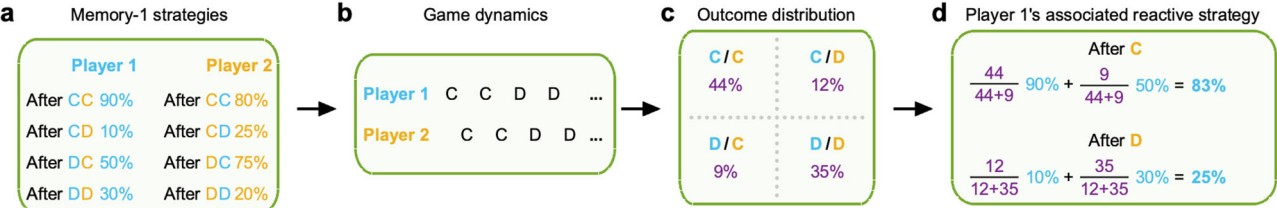

**Fig. 3 In alternating games, individuals can afford to remember less than their opponent.** We prove the following result: if two memory-1 players interact, any of the players can switch to a simpler reactive strategy (that only depends on the co-player's previous action) without changing the resulting payoffs. Here, we illustrate this result for player 1. **a** Initially, both players use memory-1 strategies. That is, a player's cooperation probability depends on the most recent decision of each player. There are four conditional cooperation probabilities. **b** The strategies determine how players interact in the alternating game. **c** Based on the strategies, we can compute how often we are to observe each pairwise outcome over the course of the game by calculating the game's stationary distribution. **d** Based on the stationary distribution, and on player 1's memory-1 strategy, we can compute an associated reactive strategy. This reactive strategy only consists of two conditional cooperation probabilities. They determine what to do if the co-player cooperated (or defected) in the previous round. The cooperation probabilities can be calculated as a weighted average of the respective memory-1 strategy's cooperation probabilities. The resulting reactive strategy for player 1 yields the same outcome distribution against player 2 as the original memory-1 strategy. We note that for this result, the assumption of alternating moves is crucial. In the simultaneous game, the respectively defined reactive strategy does not yield the same outcome distribution against player 2 as the original memory-1 strategy (see Supplementary Information).

defined by $FBF = (1, 0, 1, 1)$ and also referred to as Forgiver[27], has been successful in evolutionary competitions. Out of these examples, *ALLD* and *TFT* are reactive, whereas *WSLS* and *FBF* are not. We say a strategy is deterministic if each conditional cooperation probability is either zero or one. In particular, all of the above examples are deterministic. Otherwise, we call the strategy stochastic.

Note that our analysis includes the possibility that players sometimes make errors. That is, when a player decides to cooperate, there is some probability $\varepsilon$ that the player defects by mistake. Conversely, a player who intends to defect may cooperate with the same probability. We refer to the case of $\varepsilon = 0$ as the game without errors, and to the case of $\varepsilon > 0$ as the game with errors. We note that even a strategy that is deterministic becomes fully stochastic in the game with errors because in that case, a player's effective cooperation probability is always between $\varepsilon$ and $1 - \varepsilon$.

Considering memory-1 strategies is useful for two reasons. First, such strategies are straightforward to interpret, and the respective conditional probabilities can be easily inferred from experiments[29]. Second, when both players use memory-1 strategies, their average payoffs are simple to compute (see also Methods). To this end, suppose player 1 uses the strategy **p** and player 2 adopts strategy **q**. By representing the game as a Markov chain, we can compute the stationary distribution $\mathbf{v} = (v_{CC}, v_{CD}, v_{DC}, v_{DD})$. The entries of this stationary distribution give the probabilities of observing each of the four possible combinations of the players' actions over the course of the game. Based on this stationary distribution, we define player 1's payoff as $\pi(\mathbf{p}, \mathbf{q}) = (v_{CC} + v_{DC})b - (v_{CC} + v_{CD})c$, and similarly for player 2. While the baseline scenario focuses on memory-1 strategies, our results are more general. For example, when we describe which memory-1 strategies are Nash equilibria in the following, co-players are allowed to deviate to strategies with arbitrarily long (but finite) memory. Moreover, similar approaches can also be used to explore the evolutionary dynamics of memory-2 strategies, as we will discuss later.

**A recipe for identifying Nash equilibria for alternating games**. To predict which memory-1 strategies evolve in the alternating game, we first characterize which of them are Nash equilibria. In the following, we refer to a strategy **q** as a Nash equilibrium if $\pi(\mathbf{q}, \mathbf{q}) \geq \pi(\mathbf{p}, \mathbf{q})$ for all alternative memory-1 strategies **p** (for stronger results, see Supplementary Note 2). That is, against a co-player who adopts the Nash equilibrium strategy **q**, a player has no incentive to choose any different memory-1 strategy. The

notion of Nash equilibrium is closely related to evolutionary robustness[30]. In a population of size $N$, a resident strategy **q** is called evolutionary robust if no mutant strategy **p** has a fixation probability larger than neutral, $1/N$. When selection is sufficiently strong, strategies are evolutionary robust if and only if they are Nash equilibria[31].

Verifying that a given strategy **q** is a Nash equilibrium is not straightforward. In principle, this requires us to compare its payoff to the payoff of all possible mutant strategies **p**, taken from the uncountable set of all memory-1 strategies. However, for alternating games, it is possible to simplify the task in two steps (see Supplementary Note 2 for details). The first step is to show that it is sufficient to compare **q** to all reactive strategies, a strategy set of a lower dimension. The intuition for this result is as follows. Even if player 1 starts out with an arbitrary memory-1 strategy **p**, it is always possible to find an associated reactive strategy $\tilde{\mathbf{p}}$ that yields the same stationary distribution and the same payoff against **q** (Fig. 3). That is, to find the best response to a strategy that remembers both players' last moves, it is sufficient to explore all strategies that only remember the co-player's last move. In particular, not only is there no advantage of having a strictly larger memory than the opponent, as shown by Press and Dyson for simultaneous games[39]. A player can afford to remember strictly less in the alternating game.

The second step is to show that we do not need to consider all reactive strategies to find the best response against **q**. Instead, it suffices to consider all deterministic reactive strategies. By combining these two steps, it becomes straightforward to check whether a given memory-1 strategy is a Nash equilibrium. We only need to compare its payoff against itself to the four payoffs that can be achieved by deviating to Always Defect (*ALLD*), Always Cooperate (*ALLC*), Tit-for-Tat (*TFT*), or Anti-Tit-for-Tat (*ATFT*).

**Equilibria in alternating games without errors**. Using the above recipe, we first explore which memory-1 strategies can sustain full cooperation in games without errors (see Supplementary Note 2 for all derivations). To this end, we call a memory-1 strategy a *partner*[32,33] if (*i*) it is fully cooperative against itself, and (*ii*) if it is a Nash equilibrium (such strategies are referred to as 'good' by Akin[34–36]). We find that partners are exactly those memory-1 strategies **q** for which the following three conditions hold,

$$q_{CC} = 1, \quad q_{CD} \leq 1 - \frac{c}{b-c}q_{DD}, \quad q_{CD} \leq 1 - \frac{c}{b}q_{DC}. \tag{2}$$

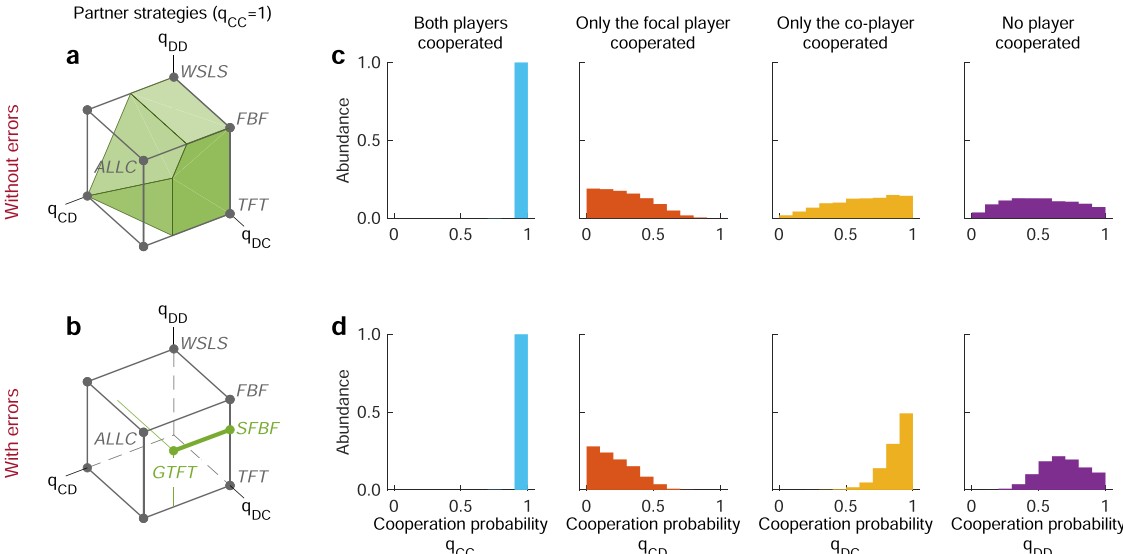

**Fig. 4 Partner strategies in alternating games with and without errors.** Partner strategies sustain cooperation in a Nash equilibrium. All such strategies are required to cooperate after mutual cooperation, such that the respective cooperation probability $q_{CC}$ is equal to one. **a** In the absence of errors, the remaining three cooperation probabilities can be chosen arbitrarily, subject to the constraints in Eq. (2). The resulting set of partner strategies takes the shape of a polyhedron. **b** In the presence of errors, this polyhedron degenerates to a single line segment. This line segment comprises all strategies between Generous Tit-for-Tat (*GTFT*) and Stochastic Firm-but-Fair (*SFBF*). **c**, **d** We compare these equilibrium results to evolutionary simulations. To this end, we record all strategies that emerge over the course of the simulation. Here, we plot the probability distribution of those strategies that yield at least 80% cooperation against themselves. Without errors, the probability distributions for $q_{CD}, q_{DC}, q_{DD}$ are comparably flat. With errors, players tend to cooperate if they exploited their opponent in the previous round, $q_{DC} \approx 1$. Moreover, they cooperate with some intermediate probability after mutual defection, $q_{DD} \approx 2/3$. Both effects are in line with previous simulation studies[25,26], and they confirm the theory. Simulations are run for $b/c = 3$, and $\varepsilon = 0$ or $\varepsilon = 0.02$. For the other parameter values and further details on the simulations, see Methods. Source data are provided as a Source Data file.

The first condition is needed to ensure that the strategy is fully cooperative against itself. The other two conditions restrict how cooperative a player is allowed to be after having been exploited by the co-player. If these last two conditions are violated, the strategy **q** can either be invaded by *ALLD* or *ATFT*. Together, the three requirements in (2) define a three-dimensional polyhedron within the space of all memory-1 strategies (Fig. 4a). The volume of this polyhedron increases with the benefit to cost ratio $b/c$. While the polyhedron never contains *ALLC*, it always contains the conditionally cooperative strategies *TFT* and *GRIM* (for these two strategies, we additionally require the respective players to cooperate in the very first round to ensure payoffs are well-defined, see Supplementary Information). Moreover, for $b \geq 2c$, the polyhedron contains *WSLS* and *FBF* (independent of the outcome of the first round).

Similarly, we can also identify all Nash equilibria where the players mutually defect. We refer to the respective strategies as *defectors*. We obtain the following necessary and sufficient conditions,

$$q_{DD} = 0, \quad q_{DC} \leq \frac{c}{b}(1 - q_{CD}), \quad q_{DC} \leq \frac{c}{b-c}(1 - q_{CC}). \quad (3)$$

Again, the first equation ensures that two players with the respective strategy end up mutually defecting against each other. The other two conditions ensure that the strategy is comparably unresponsive towards a co-player who tries to initiate cooperation. Similar to before, the three conditions define a three-dimensional polyhedron (Supplementary Fig. 2a). The set of defectors is non-empty for all parameter values, and it always contains the strategy *ALLD*.

Finally, we identify a third class of Nash equilibria, referred to as *equalizers*[43]. As in the simultaneous game[39], equalizers are strategies that unilaterally control the co-player's payoff. If one

player adopts an equalizer strategy, the co-player's payoff is fixed, independent of the co-player's strategy[44–48]. In the alternating game, these strategies are characterized by

$$q_{CD} = \frac{b\,q_{CC} - c(1 + q_{DD})}{b - c}, \quad q_{DC} = \frac{b\,q_{DD} + c(1 - q_{CC})}{b - c}. \quad (4)$$

When both players adopt an equalizer strategy, neither player has anything to gain from deviating; the resulting outcome is a Nash equilibrium.

We also show a converse result: If a memory-1 strategy for the alternating game is a Nash equilibrium, then it either needs to be a partner, a defector, or an equalizer. Remarkably, the same three strategy classes also arise as Nash equilibria of the simultaneous game[31]. Even the algebraic conditions for being a partner, defector, or equalizer coincide (however, the existing proof for the simultaneous game[31] is somewhat more intricate than the proof for the alternating game that we provide in Supplementary Note 4). There is, however, one difference. In the simultaneous game, there is a fourth class of Nash equilibria, referred to as 'alternators'[31]. Alternators cooperate in one round, only to defect in the next. In Supplementary Note 2, we show that such patterns of behavior cannot emerge among memory-1 players in the alternating game.

**Equilibria in alternating games with errors**. Next, we explore how the Nash equilibria change when we introduce errors. In the following, we discuss the case of partner strategies; the analogous results for defectors and equalizers are derived in Supplementary Note 2. For partner strategies, we find that errors impose additional constraints. First, partners only exist when errors are sufficiently rare, $\varepsilon < \frac{1}{2}\left(1 - \frac{c}{b}\right)$. Second, the respective conditions are

now considerably more restrictive,

$$q_{CC} = q_{DC} = 1, \; q_{CD} \le 1 - \frac{c}{(1 - 2\varepsilon)b},$$

$$q_{DD} = \frac{(1 - 2\varepsilon)(b + \varepsilon c\, q_{CD}) - c}{(1 - 2\varepsilon)(b + \varepsilon c)}. \quad (5)$$

In particular, if the co-player cooperated in the previous round, partners are strictly required to cooperate in the next round, independent of their own previous action (because now $p_{DC} = 1$). If the co-player defected, partners need to cooperate with a well-defined probability, as defined by the last two conditions in (5). The last condition guarantees that neither *ALLC* nor *TFT* has a selective advantage against **q**. In the game without errors, this requirement is satisfied automatically. There, all strategies with $q_{CC} = 1$ yield the full cooperation payoff $b - c$ against each other. In the game with errors, however, such strategies are no longer neutral. Instead, they differ in how quickly they are able to restore cooperation after an error, and to which extent they are able to capitalize on their co-players' mistakes. Noisy environments thus impose additional constraints on self-cooperative strategies to be stable.

As a result of these additional constraints, the three-dimensional polyhedron degenerates to a one-dimensional line segment (Fig. 4b). On one end of this line segment, there is Generous Tit-for-Tat, which also arises in the simultaneous game[6,49],

$$GTFT = \left( 1, \; 1 - \frac{c}{(1 - 2\varepsilon)b}, \; 1, \; 1 - \frac{c}{(1 - 2\varepsilon)b} \right) \quad (6)$$

On the other end of this line segment, we find a strategy that resembles the main characteristics of Firm-but-Fair[3]; we thus refer to this strategy as Stochastic Firm But Fair,

$$SFBF = \left( 1, \; 0, \; 1, \; \frac{(1 - 2\varepsilon)b - c}{(1 - 2\varepsilon)(b + \varepsilon c)} \right) \quad (7)$$

Behaviors similar to Stochastic Firm-but-Fair (*SFBF*) have been observed in early simulations of alternating games[25,26]. There, it was found that evolutionary trajectories often lead to strategies that are deterministic, except that they randomize after mutual defection. Our results provide an analytical justification: *SFBF* is the only such strategy that is a Nash equilibrium.

The above conditions in (5) provide a complete characterization of all partner strategies in the alternating game with errors. Despite decades of research, an analogous characterization for the simultaneous game is not yet available (Fig. 2). However, it is known that particular strategies, most importantly *WSLS*, can be evolutionarily stable in the presence of noise[40]. That is, in the simultaneous game, cooperation can be sustained with a simple deterministic strategy if $b > 2c$. In contrast, conditions (5) imply that no such deterministic strategy is available in the alternating game. Moreover, while the partner strategies characterized by (5) are Nash equilibria, we show in the Supplementary Information that they all are vulnerable to neutral invasion by either *ALLC* or *TFT* (in fact by all strategies with $q_{CC} = q_{DC} = 1$). These results suggest that cooperation can still evolve in alternating games, but it may be less robust than in the simultaneous game.

**Evolutionary dynamics of alternating games**. In order to test these equilibrium predictions, we next explore which behaviors emerge when the players' strategies are subject to evolution. To this end, we consider a population of $N$ players. Each member of the population is equipped with a memory-1 strategy. They obtain payoffs by interacting with all other population members. To model the spread of successful strategies, we assume individuals with high payoffs are imitated more often[50] (or

equivalently, such individuals produce more offspring[51]). In addition, new strategies are introduced through random exploration (or equivalently, through mutations). These random strategies are uniformly taken from the space of all memory-1 strategies. We capture the resulting dynamics with computer simulations. For details, see Methods.

First, we explore the evolutionary dynamics for fixed game parameters. We record which strategies the players use over the course of evolution to sustain cooperation. In Fig. 4, we represent those strategies that yield a cooperation rate against themselves of at least 80%; other threshold values lead to similar conclusions. We call these strategies "self-cooperative". By definition, players with these strategies are likely to cooperate after mutual cooperation. Here, we are thus interested in how they react when either one or both players defected. Without errors, the respective conditional cooperation probabilities show quite some variation. As a result, the distributions in Fig. 4c are comparably flat. Overall, players act in such a way that the partner conditions (2) are satisfied, but they show no preference for a particular partner strategy. Once we allow for errors, the evolving strategies change (Fig. 4d). Players tend to always cooperate if the co-player did so in the previous round, with $q_{CC} \approx q_{DC} \approx 1$. Moreover, after mutual defection, they cooperate with some strictly positive probability. Both patterns are predicted by our equilibrium conditions (5). We find a similar match between static theory and evolutionary simulations for defectors, or when we explore evolution in the simultaneous game (Supplementary Figs. 1–3).

In a next step, we compare the dynamics of the alternating and the simultaneous game across different parameter values. To this end, we systematically vary the benefit of cooperation, the population size, the selection strength, and the mutation rate (Fig. 5). In games without errors, we observe hardly any difference between the alternating and the simultaneous game. Both games yield almost identical cooperation rates over time, and these cooperation rates are similarly affected by parameter changes. A difference between the two games only becomes apparent when players need to cope with errors. Here, the simultaneous game leads to systematically higher cooperation rates than the alternating game. This difference is most visible for intermediate benefit-to-cost ratios and intermediate error rates, as one may expect: For small benefits and frequent errors, cooperation evolves in neither game, whereas for large benefits and rare errors, cooperation evolves in both games (Supplementary Fig. 4).

**Evolutionary results beyond the baseline scenario**. Our baseline scenario represents an idealized model of alternating interactions. It assumes (*i*) the game is infinitely repeated, (*ii*) players move in a strictly alternating fashion, (*iii*) games take place in a well-mixed population, and (*iv*) players use memory-1 strategies. In the following, we use simulations to explore the effect of each of these assumptions in turn. Here, we briefly summarize the respective results. For an exact description of the models, and for a more detailed discussion of the results, we refer to Supplementary Note 3.

We start by considering games with finitely many rounds. To incorporate a finite game length, we assume that each time both players have made a decision, the game continues with a constant probability $\delta$. Figure 6a–c shows the respective evolutionary results for $\delta = 0.96$ (such that games last for 25 rounds on average). We observe similar results as in the infinitely repeated game: The simultaneous game leads to more cooperation (Fig. 6a); moreover, if players cooperate, their strategies exhibit the main characteristics of *WSLS* in the simultaneous game, and of *SFBF* and *GTFT* in the alternating game (Fig. 6b). Further simulations

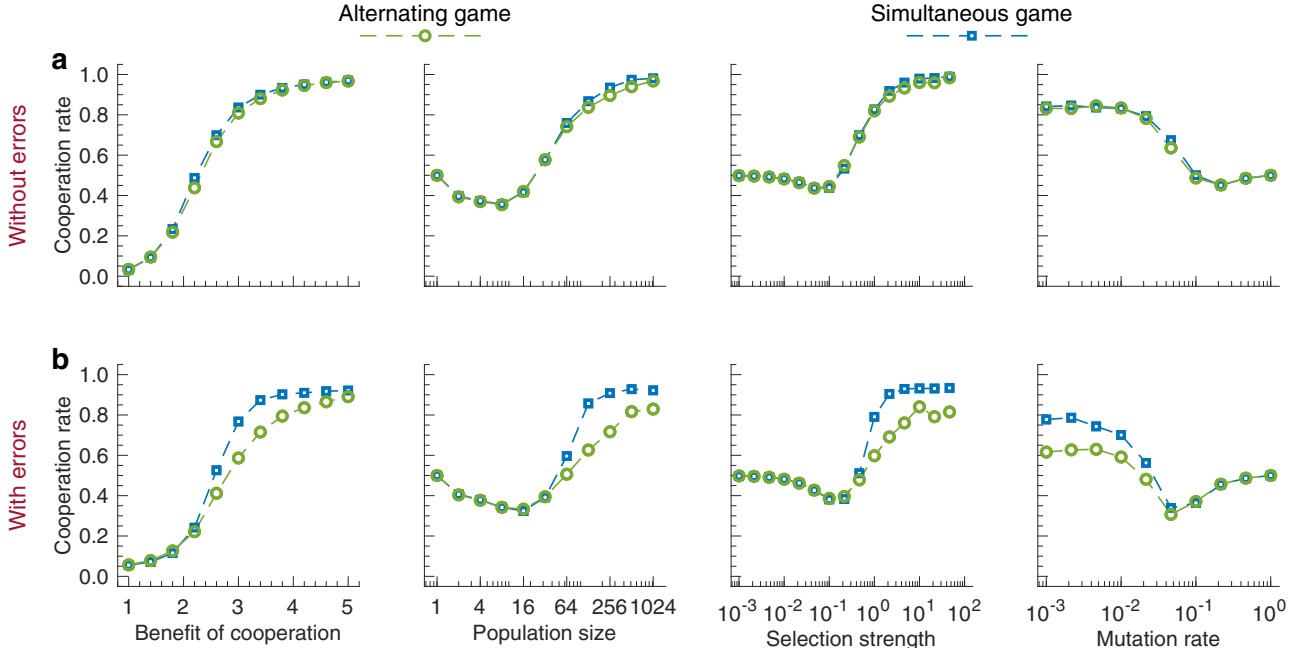

**Fig. 5 Comparing evolution in the alternating and the simultaneous game.** To compare the two game versions, we have run additional evolutionary simulations. We systematically vary the benefit of cooperation, the population size, the selection strength, and the mutation rate. In addition, we vary how likely players make errors. Either they make no errors at all ($\varepsilon = 0$), or they make errors at some intermediate rate ($\varepsilon = 0.02$). **a** In the absence of errors, there is virtually no difference between the simultaneous and the alternating game. Both games yield the same cooperation rates, and they respond to parameter changes in the same way. For the given baseline parameters, cooperation is favored for large benefits of cooperation, population sizes, and selection strengths. It is disfavored for intermediate and large mutation rates. **b** With errors, the cooperation rates in the alternating game are systematically below the simultaneous game. The lower cooperation rates are related to our analytical result that no cooperative memory-1 strategy in the alternating game is evolutionarily stable. In contrast, in the simultaneous game with errors, *WSLS* can maintain cooperation[42,53], it is evolutionarily stable[41], and it readily evolves in evolutionary simulations (Supplementary Fig. 1). As baseline parameters we use a benefit of cooperation $b = 3$, population size $N = 100$, selection strength $\beta = 1$, and the limit of rare mutations $\mu \to 0$[65,66]. Source data are provided as a Source Data file.

suggest that these qualitative results hold when players interact for at least ten rounds (Supplementary Fig. 5). When interactions are shorter, cooperation is unlikely to evolve at all (Fig. 6c).

In the next step, we explore irregular alternation patterns. To this end, we assume that every time a player has made a decision, with probability $s$ it is the other player who moves next. We refer to $s$ as the game's switching probability. For $s = 1$, we recover the baseline scenario, in which players strictly alternate. For $s = 1/2$, the player to move next is determined randomly. Simulations suggest that in both cases, players again use strategies akin to *GTFT* and *SFBF* to sustain cooperation (Fig. 6e). However, the robustness of the strategies depends on the switching probability. In particular, mutual cooperation is most likely to evolve when players alternate regularly (Fig. 6f, Supplementary Fig. 6).

To explore the effect of population structure, we follow the framework of Brauchli et al.[52]. Instead of well-mixed populations, players are now arranged on a two-dimensional lattice. They use memory-1 strategies to engage in pairwise interactions with each of their neighbors. For the simultaneous game, we recover the main results of Brauchli et al.[52]: population structure can further enhance cooperation, and it makes it more likely that strategies similar to *WSLS* evolve (Fig. 6g–i). For the alternating game, we observe that cooperation remains the most abundant outcome, but the spatial structure does not necessarily result in homogeneous populations any longer. Instead, in some simulations, we find cooperative and non-cooperative strategies to stably coexist (one particular instance is shown in Fig. 6h).

Finally, we also analyzed the impact of a larger memory. Exploring the dynamics among general memory-$k$ strategies is not straightforward, as the strategy space increases rapidly. For instance, while there are only 16 pure memory-1 strategies, there

are 65,536 memory-2 strategies and more than $10^{19}$ memory-3 strategies[41]. We thus confine ourselves to pure memory-2 strategies in the following. In a first step, we explored which of these strategies are evolutionarily stable, see Supplementary Fig. 7a. For the simultaneous game, we find many such strategies, including several strategies with high cooperation rates. In the alternating game, we only find one strategy that is evolutionarily stable for a wide range of parameters, *ALLD*. Nevertheless, with respect to the evolving cooperation rates, stochastic evolutionary trajectories hardly show any difference between alternating and simultaneous games. The two games differ, however, in terms of the strategies that evolve, and in how robust these strategies are (Supplementary Fig. 7b–e).

## Discussion
An overwhelming majority of past research on reciprocity deals with repeated games where individuals simultaneously decide whether to cooperate[3,18]. In contrast, most natural occurrences of reciprocity require asynchronous acts of giving. Cooperation routinely takes the form of assisting a peer, providing a gift, or taking the lead in a joint endeavor[22–24]. In such examples, simultaneous cooperation can be unfeasible, undesirable, or unnecessary. Herein, we have thus explored which strategies arise in alternating games where individuals make their decisions in turns. In such games, one individual's cooperation is reciprocated not immediately, but at some point in the future.

To explore the dynamics of cooperation in alternating games, we first describe all Nash equilibria among the memory-1 strategies. Memory-1 strategies are classical tools that have been used to describe the evolutionary dynamics of repeated games for

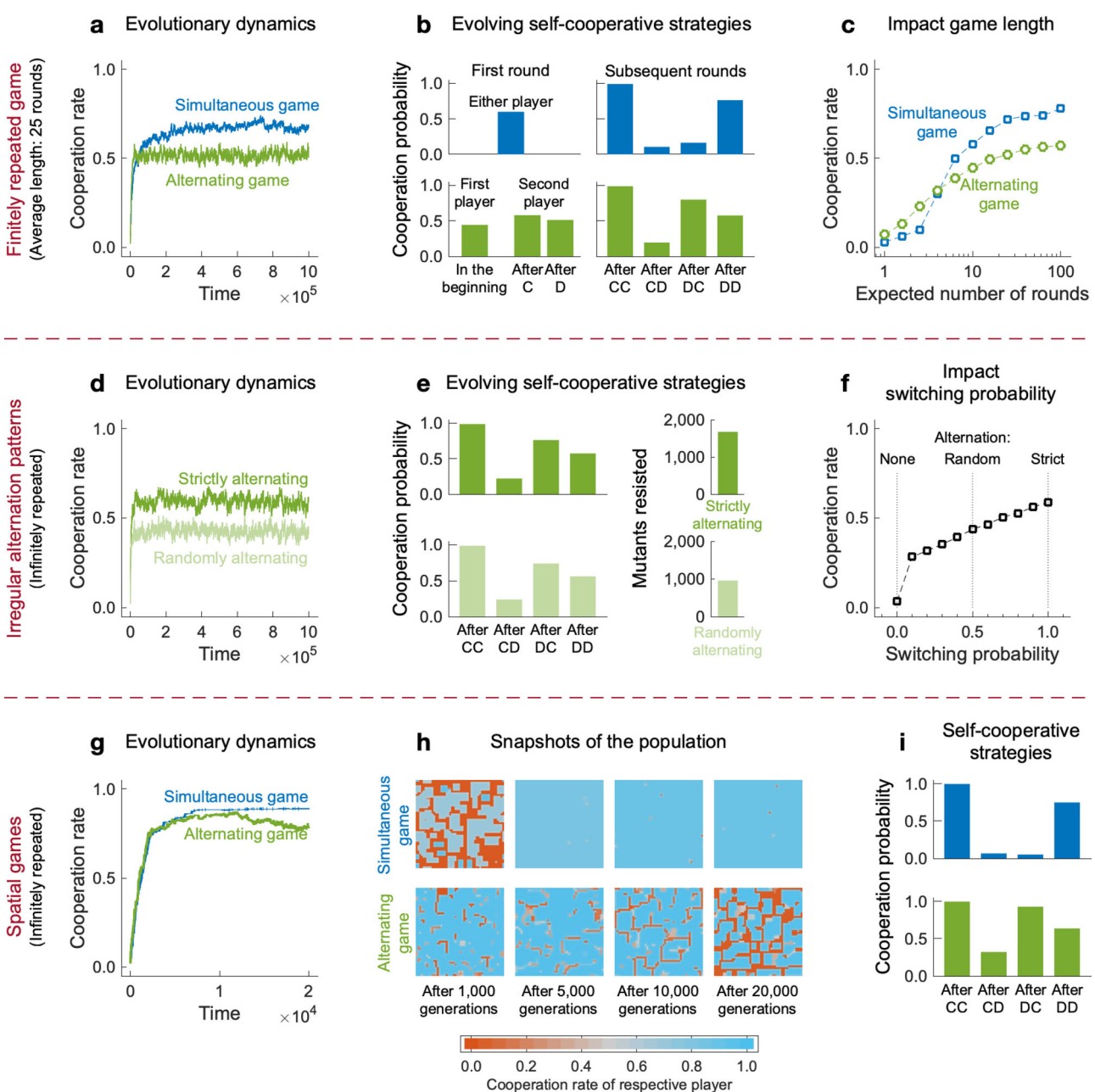

**Fig. 6 Robustness of evolutionary results.** We have explored the robustness of our results with various model extensions. Here, we display results for three of them, illustrating the impact of finitely repeated games, of irregular alternating patterns, and of population structure. **a–c** The baseline model assumes infinitely repeated games; here we show simulations for games with a finite expected length. If there are sufficiently many rounds, the simultaneous game again leads to more cooperation than the alternating game, and the evolving strategies are largely similar to the ones observed in the baseline model. **d–f** The baseline model assumes that players move in a strictly alternating fashion. Instead, here we assume that after each player's move, the other player moves with some switching probability $s$. The case $s = 1$ corresponds to strict alternation, whereas $s = 1/2$ represents a case in which the next player to move is completely random. We observe that irregular alternation patterns hardly affect which strategies players use to cooperate. However, it affects the robustness of these strategies. Overall, cooperation is most likely to evolve under strict alternation. **g–i** Finally, instead of well-mixed populations, we consider games on a lattice. For the given parameter values, we observe that simultaneous games eventually lead to homogeneous cooperative populations. While this outcome is also possible for alternating games, some simulations also lead to the coexistence of cooperators and defectors (shown here in panel (**h**)). The evolving self-cooperative strategies are similar to the strategies that evolve in the baseline model. For a detailed description of these simulations, see Methods and Supplementary Information. Source data for panels **a–f**, **i** are provided as a Source Data file.

several decades[25,42,53]. However, most of the early work on memory-1 strategies was restricted to evolutionary simulations. Only with the pioneering work of Press and Dyson[39] and others[30–38], better mathematical techniques have become available. Using these techniques, it has become possible to describe all Nash equilibria of the infinitely repeated simultaneous game

without errors[31]. Herein, we make similar progress for the alternating game, both for the case with and without errors (for the simultaneous game with errors, a complete characterization of the Nash equilibria remains an open problem, see Fig. 2).

Our results suggest that there are both unexpected parallels and important differences between simultaneous and alternating

games. The parallels arise when individuals do not make errors. Here, the two models of reciprocity make the same predictions about the feasibility of cooperation. Cooperation evolves in the same environments, and it can be maintained using the same strategies. However, once individuals make mistakes, the predictions of the two models diverge. First, the two models require different kinds of strategies to maintain cooperation. In the simultaneous game, cooperation can be sustained with the deterministic memory-1 strategy Win-Stay Lose-Shift[42,53]. Individuals with that strategy simply reiterate their previous behavior if it was successful, and they switch their behavior otherwise. In contrast, in the alternating game, no simple deterministic rules for cooperation exist. Although there are still infinitely many memory-1 strategies that can maintain cooperation, all of them require individuals to randomize occasionally. One example of such a strategy for alternating games is *SFBF*. Individuals with this strategy always reciprocate a co-player's cooperation, never tolerate exploitation, and cooperate with some intermediate probability if both players defected. Similar behaviors have been observed in earlier simulations[25,26]. Our results provide a theoretical underpinning: *SFBF* is the unique memory-1 strategy that can sustain cooperation while retaliating against unconditional defectors in the strongest possible way.

The simultaneous game and the alternating game also differ in how stable cooperation is in evolving populations. In the simultaneous game, the evolution of cooperation is hardly affected by errors, provided the error rate is below a certain threshold (Fig. 5, Supplementary Fig. 4). In some instances, errors can even enhance cooperation[54]. This body of work is based on the insight that evolutionarily stable cooperation is impossible in simultaneous games without errors[55–59]. For any cooperative resident, it is always possible to find neutral mutant strategies that eventually lead to the demise of cooperation. However, once individuals occasionally commit errors, a strategy like *WSLS* is no longer neutral with respect to other cooperative strategies; it becomes evolutionarily stable[40,56]. The situation is different in alternating games. Even in the presence of rare errors, strategies like *SFBF* remain vulnerable. They can be invaded by unconditional cooperators or by any other strategy that fully reciprocates a co-player's cooperation.

Despite these differences in the stability of their main strategies, evolving cooperation rates in the simultaneous and the alternating game are often surprisingly similar. To interpret these results, we note that when evolution is stochastic and takes place in finite populations, no strategy persists indefinitely. Even evolutionarily stable strategies are invaded eventually. As a result, the overall abundance of cooperation is not only determined by the stability of any given strategy. Instead, it depends on additional aspects, such as the time it takes cooperative strategies to reappear when they are invaded. The relative importance of these different aspects depends on the details of the considered evolutionary process. To further illustrate these observations, we have run additional simulations for memory-1 players with local mutations[60] (see Supplementary Note 3). Because evolutionary stability considerations are less relevant when mutations are local, we observe that the cooperation rates of the alternating and the simultaneous game become more similar (Supplementary Fig. 8).

Cooperation is defined as a behavior where individuals pay a cost in order to increase the payoff or fitness of someone else[2]. When individuals interact repeatedly, such cooperative interactions can be maintained by reciprocity. Here, we have argued that in many examples, reciprocity arises as a series of asynchronous acts of cooperation. Most often, people do favors not to be rewarded immediately, but to request similar favors in the future. Such consecutive acts of cooperation also appear to be at work when vampire bats[20], sticklebacks[23], ibis[24], tree swallows[61], or

macaques[62] engage in reciprocity. We have shown that mutual cooperation is still possible in such alternating exchanges. But compared to the predominant model of reciprocity in simultaneous games, cooperation requires different kinds of strategies, and it is more volatile.

## Methods

**Calculation of payoffs**. When two players with memory-1 strategies interact, their expected payoffs can be computed by representing the game as a Markov chain[3]. To this end, suppose the first player's strategy is $\mathbf{p} = (p_{CC}, p_{CD}, p_{DC}, p_{DD})$, and the second player's strategy is $\mathbf{q} = (q_{CC}, q_{CD}, q_{DC}, q_{DD})$. Depending on the most recent actions of the two players (which can be either $CC$, $CD$, $DC$, or $DD$), we can compute how likely we are to observe each of the four outcomes in the following round. For the alternating game, we obtain the following transition matrix[25],

$$M_A = \begin{pmatrix} p_{CC}q_{CC} & p_{CC}(1-q_{CC}) & (1-p_{CC})q_{CD} & (1-p_{CC})(1-q_{CD}) \\ p_{CD}q_{DC} & p_{CD}(1-q_{DC}) & (1-p_{CD})q_{DD} & (1-p_{CD})(1-q_{DD}) \\ p_{DC}q_{CC} & p_{DC}(1-q_{CC}) & (1-p_{DC})q_{CD} & (1-p_{DC})(1-q_{CD}) \\ p_{DD}q_{DC} & p_{DD}(1-q_{DC}) & (1-p_{DD})q_{DD} & (1-p_{DD})(1-q_{DD}) \end{pmatrix}.$$
(8)

Based on this transition matrix, we compute how often players observe each of the four outcomes. To this end, we solve the equation for the stationary distribution, $\mathbf{v} = \mathbf{v}M_A$. In most cases, the solution of this equation is unique. Uniqueness is guaranteed, for example, when the players' strategies $\mathbf{p}$ and $\mathbf{q}$ are fully stochastic, or when the error rate is positive. In exceptional cases, however, the transition matrix can allow for two or more stationary distributions. In that case, the outcome of the game is still well-defined, after specifying how players act in the very first round.

Given the stationary distribution $\mathbf{v} = (v_{CC}, v_{CD}, v_{DC}, v_{DD})$, we define the players' payoffs as

$$\begin{aligned} \pi_1 &= (v_{CC} + v_{DC})b - (v_{CC} + v_{CD})c, \\ \pi_2 &= (v_{CC} + v_{CD})b - (v_{CC} + v_{DC})c. \end{aligned}$$
(9)

This definition implicitly assumes that the game is indefinitely repeated and that future payoffs are not discounted. However, analogous formulas can be given in case there is a constant continuation probability $\delta$, or equivalently if future payoffs are discounted by $\delta$ (see Supplementary Note 3).

We compare our results for the alternating game with the corresponding results for the standard repeated prisoner's dilemma, where players decide simultaneously. Payoffs for the simultaneous game can be calculated in the same way as before. Only the transition matrix needs to be replaced by[3]

$$M_S = \begin{pmatrix} p_{CC}q_{CC} & p_{CC}(1-q_{CC}) & (1-p_{CC})q_{CC} & (1-p_{CC})(1-q_{CC}) \\ p_{CD}q_{DC} & p_{CD}(1-q_{DC}) & (1-p_{CD})q_{DC} & (1-p_{CD})(1-q_{DC}) \\ p_{DC}q_{CD} & p_{DC}(1-q_{CD}) & (1-p_{DC})q_{CD} & (1-p_{DC})(1-q_{CD}) \\ p_{DD}q_{DD} & p_{DD}(1-q_{DD}) & (1-p_{DD})q_{DD} & (1-p_{DD})(1-q_{DD}) \end{pmatrix}.$$
(10)

Although the two matrices share many similarities, the resulting dynamics can be very different. For example, if the two players use *TFT*, then the matrix $M_S$ allows for three invariant sets (corresponding to mutual cooperation, mutual defection, and alternating cooperation). However, the respective matrix $M_A$ only allows for the first two invariant sets[25]. More generally, $M_S$ allows for equilibria where players cooperate in one round but defect in the next round. Such equilibria are impossible for $M_A$ (see Supplementary Note 2).

We sometimes assume players commit errors. We incorporate errors by assuming that with probability $\varepsilon$, a player who intends to cooperate defects by mistake. Analogously, a player who wishes to defect cooperates instead with the same probability. Such errors are straightforward to incorporate into the model. For $\varepsilon > 0$, a player's strategy $\mathbf{p}$ translates into an effective strategy $\mathbf{p}^\varepsilon := (1-\varepsilon)\mathbf{p} + \varepsilon(\mathbf{1} - \mathbf{p})$. To compute the payoffs of strategy $\mathbf{p}$ against strategy $\mathbf{q}$ in the presence of errors, we apply the formulas (8)–(10) to the strategies $\mathbf{p}^\varepsilon$ and $\mathbf{q}^\varepsilon$.

**Evolutionary dynamics**. In the following, we describe the evolutionary process for the baseline scenario. For the various model extensions (Fig. 6, Supplementary Fig. 5–Supplementary Fig. 8), we use appropriately adapted versions of this process, as described in more detail in Supplementary Note 3. To model how successful strategies spread in well-mixed populations, we use a pairwise comparison process[50]. This process considers a population of constant size $N$. Initially, all population members are unconditional defectors. Each player derives a payoff by interacting with all other population members; for each pairwise interaction, payoffs are given by Eq. (9).

To model how strategies with a high payoff spread within a population, we consider a model in discrete time. In each time step, one player is chosen from the population at random. This player is then given an opportunity to revise its strategy. The player can do so in two ways. First, with probability $\mu$ (the mutation rate), the player may engage in random strategy exploration. In this case, the player discards its strategy and samples a new strategy uniformly at random from the set

of all memory-1 strategies. Second, with probability $1 - \mu$, the player considers imitating one of its peers. In this case, the player selects a random role model from the population. If the role model's payoff is $\pi_R$ and the focal player's payoff is $\pi_F$, then imitation occurs with a probability given by the Fermi function[63]

$$\rho = \frac{1}{1 + \exp\left[-\beta(\pi_R - \pi_F)\right]}. \tag{11}$$

If imitation occurs, the focal player discards its previous strategy and adopts the role model's strategy instead. In the formula for the imitation probability, the parameter $\beta \geq 0$ is called the strength of selection. It measures the extent to which players are guided by payoff differences between the players' strategies. For $\beta = 0$, any payoff differences are irrelevant. The focal player adopts the role model's strategy with a probability of 1/2. As $\beta$ becomes larger, payoff differences become increasingly important. In the limiting case $\beta \to \infty$, imitation only occurs if the role model's payoff at least matches the focal player's payoff.

Overall, the two mechanisms of random strategy exploration and directed strategy imitation give rise to a stochastic process on the space of all population compositions. For positive mutation rates, this process is ergodic. In particular, the average cooperation rate (as a function of the number of time steps) converges, and it is independent of the considered initial population. Herein, we have explored this process with computer simulations. We have recorded which strategies the players adopt over time and how often they cooperate on average. For most of these simulations, we assume that mutations are sufficiently rare[64]. For those simulations, we require mutant strategies to either fix in the population or to go extinct before the next mutation occurs. Under this regime, the mutant's fixation probability can be computed explicitly[9]. This in turn allows us to simulate the evolutionary dynamics more efficiently[65,66].

**Parameters and specific procedures used for the figures**. For the simulations in well-mixed populations, we used the following baseline parameters

| | |
|---|---|
| Benefit of cooperation : | $b = 3$ |
| Cost of cooperation : | $c = 1$ |
| Population size : | $N = 100$ |
| Selection strength : | $\beta = 5$ (Fig. 4, Supplementary Figures 1−3) and $\beta = 1$ (all other figures) |
| Error rate : | $\varepsilon = 0$ (without errors), or $\varepsilon = 0.02$ (with errors) |
| Mutation rate : | $\mu \to 0.$ |

$$\tag{12}$$

Changes in these parameters are systematically explored in Fig. 5 and Supplementary Fig. 4. For Figs. 4, 5, and Supplementary Fig. 1–Supplementary Fig. 6, the respective simulations are run for at least $10^7$-time steps each (measured in a number of introduced mutant strategies over the course of a simulation). For Fig. 6, Supplementary Fig. 7, and Supplementary Fig. 8, simulations are run for a shorter time (as illustrated in the respective panels that illustrate the resulting dynamics). However, here all results are obtained by averaging over 50–200 independent simulations.

To report which strategies the players use to sustain cooperation (or defection), we record all strategies that arise during a simulation that have a cooperation rate against themselves of at least 80% (in the case of self-cooperators), or a cooperation rate of less than 20% (in the case of self-defectors). In Fig. 4, Supplementary Fig. 1–Supplementary Fig. 3, and Supplementary Fig. 5, we show the marginal distributions of all strategies that we have obtained in this way. For these distributions, each strategy is weighted by how long the strategy has been present in the population. In Fig. 6, Supplementary Fig. 7, and Supplementary Fig. 8, we represent the self-cooperative strategies by computing the average of the respective marginal distributions. In some cases (Fig. 6e, Supplementary Fig. 7, Supplementary Fig. 8), we also report how robust self-cooperative strategies are on average. To this end, we record for each self-cooperative resident strategy how many mutants need to be introduced into the population until a mutant strategy reaches fixation. We consider self-cooperative strategies that resist invasion by many mutant strategies as more robust.

Finally, for the simulations for spatial populations (Fig. 6g–i), we closely follow the setup of Brauchli et al.[52]. Here, we consider a population of size $N = 2500$. Players are arranged on a $50 \times 50$ lattices with periodic boundary conditions. Players use memory-1 strategies (initially they adopt the strategy ALLD). In every generation, every player interacts in a pairwise game with each of its eight immediate neighbors. After these interactions, all players are independently given an opportunity to update their strategies. With probability $\mu = 0.002$, an updating player chooses a random strategy, uniformly taken from all memory-1 strategies (global mutations). With probability $1 - \mu$, the updating player adopts the strategy of the neighbor with the highest payoff (but only if this neighbor's payoff is better than the focal player's payoff). This elementary process is then repeated for 20,000 generations. Figure 6g, i shows averages across 50 independent simulations of the process. Figure 6h illustrates two particular realizations.

**Reporting summary**. Further information on research design is available in the Nature Research Reporting Summary linked to this article.

## Data availability
Source data for Fig. 4, Fig. 5, Fig. 6a–f and i are provided with this paper. Moreover, the raw data generated with the computer simulations, including the data that is necessary to create all figures are available online[67], at osf.io: https://doi.org/10.17605/osf.io/v5hgd. Source data are provided with this paper.

## Code availability
All simulations were performed with MATLAB_R2019b. The respective code is available online[67], at osf.io: https://doi.org/10.17605/osf.io/v5hgd.

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

## Acknowledgements

P.S.P. is supported by the National Science Foundation through the Graduate Research Fellowship Program (grant number DGE1745303), by the Centre for Effective Altruism through the Global Priorities Fellowship, and by Harvard University through graduate student fellowships. C.H. acknowledges generous funding by the Max Planck Society and by the ERC Starting Grant E-DIRECT (850529).

## Author contributions

P.S.P., M.A.N., and C.H: designed the research; P.S.P., M.A.N., and C.H: performed the research; P.S.P., M.A.N., and C.H: wrote the paper.

## Funding

## Competing interests

The authors declare no competing interests.
