## [Peer Review File · Nature Communications]

Cooperation in alternating interactions with memory constraintsREVIEWER COMMENTS

Reviewer #1 (Remarks to the Author):

See attachment [INCLUDED BELOW]

Reviewer #2 (Remarks to the Author):

Park and colleagues studied alternating cooperation games. They rightly point out that we have not sufficiently understood what would be an optimal cooperation strategy in such games. It is therefore important to study these games, the conditions that would allow robust cooperation to evolve, how these cooperation strategies look like, and how they relate to the ones of simultaneous cooperation games. However, I find Park et al.' analyses too limited and the insight provided by the MS too small. To explain:

The authors state in the abstract that "... many manifestations of reciprocity in nature are asynchronous", and that they "... explore such alternating games". This is a great promise, because it would indeed be important to understand real-life asynchronous games. However, the authors then focus on the special case of strictly alternating interactions with constant gains and losses in endless games. Even if we ignore the last condition (endless game), these are still games that are rare in real life. Moreover, the authors allowed for only memory1 strategies (remembering only the last decision of both, the player and the opponent) or reactive strategies based on only the opponent's last decision. Both, the setting and the restricted memory of players were the basis of great work published in the early 1990s by Nowak and others. And obviously, a thorough game-theoretical analysis of cooperation strategies has to start with simplifying assumptions before complexity can be added step by step. But 25 years after these ground-breaking papers, analyses are limited if they are still based on these unrealistic assumptions. Therefore, a title like "Robust cooperation in alternating interactions" is promising too much.

An important take-home message from earlier work (beginning in the early 1990s) was that probabilistic strategies are more powerful than deterministic ones. Indeed, it was immediately clear that the highly praised win-stay-loose-shift (WSLS) strategy has little relevance in people's social lives when played as a deterministic strategy. A player of deterministic WSLS would always cooperate (at a rate of 100%) after mutual defection. Humans don't. If they are forgiving and cooperative after mutual defection, it is only with a probability of < 100%. There are other arguments in favor of the strength of probabilistic strategies, much of which has already been elaborated in earlier work. Thus, when Park et al.'s abstract ends with the emphasis that "... mutual cooperation ... required probabilistic strategies", the novelty value of the paper is clearly limited. We have known this for over two decades.

Reviewer #3 (Remarks to the Author):

This paper considers the evolution of cooperation through direct reciprocity with asynchronous interactions. Understanding the evolution of cooperation is a fundamental problem in evolutionary theory and most of the work that has studied this topic using direct reciprocity has focused on simultaneously interactions. Since many real-world interactions are not simultaneous it is important to consider direct reciprocity with alternating (or more general patterns of) interactions. The current paper therefore fills an important gap in the existing literature. This paper provides a carefully and convincing analysis of the evolution of cooperation through direct reciprocity with alternating interactions that significantly extends the previously known results. The paper is well written and thoughtfully organized, and contains a number of new and interesting results that contribute meaningfully to the field. I only have two minor comments.

1. The authors assume, as is common in the field, that there are infinitely many interaction rounds in the game (and that payoffs from different rounds are not discounted). Since one of the most obvious

facts about real-world interactions is that they often consist of quite a small number of interactions (and/or have payoffs that are quite heavily discounted) it would be interesting to know if the authors have any results (even if only numerical) indicating how applicable the results of the current paper are to this situation.

2. It is known for the simultaneous game that structured populations can significantly effect the evolution of cooperation (see for example, Brauchli et al., *Evolution of Cooperation in Spatially Structured Populations*, *JTB* 200, 1999, 405-417). It would be very interesting to know what the corresponding effects of population structure would be for the alternating case that the authors are considering in the present paper.

Reviewer #1 Attachment:

“Robust cooperation in alternating interactions” by Park et. al presents a comprehensive analysis of all the Nash equilibria among memory-1 strategies in the iterated donation game, for alternating games with errors. This is a significant generalization of previous results and is a valuable contribution to the field. It will be of great interest to the wide readership of Nature Communications, and I therefore support ultimate publication. Below I have offered some suggestions that I hope the authors will take into account in a revision.

The really nice thing about this paper is that the authors bring together results for alternating and simultaneous games with and without errors. In particular, the presence (or not) of errors has led different authors to strikingly different conclusions about the the existence of evolutionary stable strategies in the Iterated Prisoner’s Dilemma (IPD) and therefore, one assumes, the evolutionary dynamics of cooperation. In particular authors such as Boyd (reference 52) and Lorberbaum et al (reference 35) have shown that specific strategies *are* evolutionary stable in the simultaneous game provided errors are nonzero but “sufficiently” rare, while it has been repeatedly shown that no strategies are evolutionary stable in when errors become vanishingly small. Similarly, the authors of this paper show (as summarized in Table 1) that in the alternating game the cooperative Nash equilibria are of a different character when errors are present vs when they are not. Yet the amount of cooperation that evolves with and without errors (Figure 4) is not notably different, in the two cases - there are some quantitative differences (Figure S4) but the shapes of the curves in Figure 4 remain the same in the two cases. They do show that the strategy distribution (Figure 3) in the alternating game is somewhat different under global mutations but I believe this warrants further discussion/explanation.

To illustrate why, I will discuss the evolutionary stability of WSLS in the simultaneous game. While it is true that WSLS is an ESS in the simultaneous game with errors, a rare mutant that causes a deviation to WSLS in p_{cd} , p_{dc} or p_{dd} by an amount Δ compared to the resident population will receive a payoff reduction of $O(\epsilon \times \Delta)$ meaning that if $\Delta \sim \epsilon$ the payoff deviation is $O(\epsilon^2)$, which terms are typically regarded as negligible when addressing the impact of errors on the IPD (see eg 52). To be a little more precise, if a population is finite then unless the population size $N > 1/\epsilon^2$ such deviations will have little effect on evolutionary dynamics. In Figure 4 the authors use $N \leq 1024$ and $\epsilon = 0.02$ which gives $1/\epsilon^2 = 2500$ i.e we expect WSLS to be invaded nearly neutrally by mutations of size $\Delta \sim \epsilon$. Such cases might be pretty rare if mutations are global, as they assume, but not if they are local. This issue is widely neglected in discussions of errors in the IPD, such as 35 and 52, where authors insist that errors are “realistic” but then only consider their effect on evolutionary dynamics in infinite populations. Since this paper explicitly discusses evolutionary dynamics in finite populations with and without errors it would be really useful to provide readers with intuitions and analysis of how the apparently contradictory sets of results in Table 1 link up.

Best wishes,

Alex Stewart

First of all, we would like to thank the editor and the reviewers for their efforts. The reviewers made many valuable suggestions how to further improve our paper, and to explore various model extensions. In the meanwhile, we have incorporated all suggestions. In particular, we have run additional simulations to explore the following aspects

- The effect of local mutations on cooperation and evolutionary stability (**Fig. S8**), as suggested by reviewer #1
- The dynamics of finitely repeated games (**Fig. 5a-c** and **Fig. S5**), as suggested by reviewers #2 and #3
- The effect of irregular alternation patterns (**Fig. 5d-f** and **Fig. S6**), as suggested by reviewer #2
- The evolution of cooperation among memory-2 strategies (**Fig. S7**), as suggested by reviewer #2
- The impact of population structure (**Fig. 5g-i**), as suggested by reviewer #3

All of the corresponding results are briefly discussed in the main text. The corresponding methods are described in detail in the new **SI Section 3: Extensions of the baseline model**. In addition to these major additions, we have also made several smaller changes related to the discussion and interpretation of our results. For a more detailed description of these changes, please find the point-by-point reply below.

We would like to thank the reviewers for prompting these changes. We believe that with these changes, the manuscript has improved substantially.

Reviewer #1.

“Robust cooperation in alternating interactions” by Park et. al presents a comprehensive analysis of all the Nash equilibria among memory-1 strategies in the iterated donation game, for alternating games with errors. This is a significant generalization of previous results and is a valuable contribution to the field. It will be of great interest to the wide readership of Nature Communications, and I therefore support ultimate publication.

Reply: Thank you for the positive assessment!

Below I have offered some suggestions that I hope the authors will take into account in a revision.

The really nice thing about this paper is that the authors bring together results for alternating and simultaneous games with and without errors. In particular, the presence (or not) of errors has led different authors to strikingly different conclusions about the the existence of evolutionary stable strategies in the Iterated Prisoner’s Dilemma (IPD) and therefore, one assumes, the evolutionary dynamics of cooperation. In particular authors such as Boyd (reference 52) and Lorberbaum et al (reference 35) have shown that specific strategies are evolutionary stable in the simultaneous game provided errors are nonzero but “sufficiently” rare, while it has been repeatedly shown that no strategies are evolutionary stable in when errors become

vanishingly small. Similarly, the authors of this paper show (as summarized in Table 1) that in the alternating game the cooperative Nash equilibria are of a different character when errors are present vs when they are not. Yet the amount of cooperation that evolves with and without errors (Figure 4) is not notably different, in the two cases - there are some quantitative differences (Figure S4) but the shapes of the curves in Figure 4 remain the same in the two cases. They do show that the strategy distribution (Figure 3) in the alternating game is somewhat different under global mutations but I believe this warrants further discussion/explanation.

Reply: That's an excellent point. In the past, the static concept of evolutionary stability has led to quite some debate in the literature on direct reciprocity. Some early papers suggest that evolutionarily stable cooperation is impossible in the iterated prisoner's dilemma. However, subsequent papers (including the references 35 and 52 mentioned above) have found that evolutionary stability is possible in games with errors.

We show that even in the presence of errors, the alternating game does not allow for evolutionarily stable cooperation. In contrast, the simultaneous game admits such an evolutionarily stable strategy, Win-Stay Lose-Shift (WSLS). One might thus expect that simultaneous games are more conducive to the evolution of cooperation. While our simulations show that simultaneous games often enjoy *some* advantage, the difference between cooperation rates is perhaps smaller than one might expect based on static considerations alone.

Of course, the stability of cooperative strategies is only one aspect that affects how much cooperation we would expect in evolutionary simulations. Another aspect, for example, is how easy cooperative strategies can invade non-cooperative ones. The relative weight of each of these aspects depends on the exact setup of the simulation. As the reviewer correctly points out below, evolutionary stability has more predictive power when populations are large and/or selection is strong.

To illustrate why, I will discuss the evolutionary stability of WSLS in the simultaneous game. While it is true that WSLS is an ESS in the simultaneous game with errors, a rare mutant that causes a deviation to WSLS in p_{cd} , p_{dc} or p_{dd} by an amount Δ compared to the resident population will receive a payoff reduction of $O(\epsilon \times \Delta)$ meaning that if $\Delta \sim \epsilon$ the payoff deviation is $O(\epsilon^2)$, which terms are typically regarded as negligible when addressing the impact of errors on the IPD (see eg 52). To be a little more precise, if a population is finite then unless the population size $N > 1/\epsilon^2$ such deviations will have little effect on evolutionary dynamics. In Figure 4 the authors use $N \leq 1024$ and $\epsilon = 0.02$ which gives $1/\epsilon^2 = 2500$ i.e we expect WSLS to be invaded nearly neutrally by mutations of size $\Delta \sim \epsilon$. Such cases might be pretty rare if mutations are global, as they assume, but not if they are local. This issue is widely neglected in discussions of errors in the IPD, such as 35 and 52, where authors insist that errors are "realistic" but then only consider their effect on evolutionary dynamics in infinite populations. Since this paper explicitly discusses evolutionary dynamics in finite populations with and without errors it would be really useful to provide readers with intuitions and analysis of how the apparently contradictory sets of results in Table 1 link up.

Reply and changes: Thank you, this is a great suggestion. To address this comment, we implemented two changes.

First, to illustrate the above considerations, we have run additional simulations for a process in which mutations are local. The setup is explained in detail in the new **SI Section 3: Extensions of the baseline model**. The corresponding results are shown as **Fig. S8**. As expected by the reviewer, local mutations reduce the importance of stability considerations. For example, in the simultaneous game with global mutations, it takes on average more than 6,000 random mutant strategies to invade a typical cooperative resident population. With local mutations, this number is reduced to 260 mutants (this number is of the order of the population size, $N = 100$, as one might expect for nearly neutral evolution).

We see a similar reduction for the alternating game. However, because cooperative strategies in the alternating game are generally less robust, the impact of local mutations is less pronounced.

Second, as suggested by the reviewer, we now provide a more detailed explanation of static equilibrium concepts (such as evolutionary stability), and their role in predicting the evolution of cooperation. This discussion follows the arguments above.

Reviewer #2.

Park and colleagues studied alternating cooperation games. They rightly point out that we have not sufficiently understood what would be an optimal cooperation strategy in such games. It is therefore important to study these games, the conditions that would allow robust cooperation to evolve, how these cooperation strategies look like, and how they relate to the ones of simultaneous cooperation games. However, I find Park et al.'s analyses too limited and the insight provided by the MS too small.

Reply: We are happy to learn that the reviewer finds our general research question interesting and relevant. Also the reviewer's criticisms regarding the scope of our analysis are well taken. In the meanwhile, we have addressed these criticisms with several new analyses and further computer simulations. As a result, we believe the quality of the manuscript has improved considerably. In particular, we are now better able to discuss how our earlier findings extend to more general environments. Please find more details below.

To explain:

The authors state in the abstract that "... many manifestations of reciprocity in nature are asynchronous", and that they "... explore such alternating games". This is a great promise, because it would indeed be important to understand real-life asynchronous games. However, the authors then focus on the special case of strictly alternating interactions with constant gains and losses in endless games. Even if we ignore the last condition (endless game), these are still games that are rare in real life.

Reply: To analyze cooperation in alternating games, we made a number of simplifying assumptions. One assumption is that the game is infinitely repeated. Another assumption is that players move in a strictly alternating fashion. Of course, both assumptions are mathematical idealizations. We fully agree with the reviewer that to increase the external validity of our results, we should explore what happens when these assumptions are no longer met. For the revised manuscript, we have done exactly this. We have run extensive computer simulations

addressing both, finitely repeated games and irregular alternation patterns. We describe the corresponding results further below.

Before that, let us briefly explain why we focused on infinitely repeated games in the first place. There are three major reasons why we consider this analysis to be useful.

- (i) **It allows us to better compare our results on alternating games to previous work on simultaneous games.** One of our key aims is to compare the dynamics of the alternating game to the simultaneous game. However, most of the corresponding work on the simultaneous game assumes an infinite game length. In particular, the memory-1 Nash equilibria of the simultaneous game are only known in the infinitely repeated case (Stewart & Plotkin, PNAS 2014).
- (ii) **It allows us to consider a simpler strategy space.** In infinitely repeated games, the impact of a player's first round behavior is typically negligible. As a result, the players' strategies take a simpler form. Instead of a 5-dimensional vector (in the simultaneous game) or a 7-dimensional vector (in the alternating game), strategies for the finitely repeated game are only 4-dimensional (for both the simultaneous and the alternating game). This reduced dimensionality does not only simplify the mathematics. It also makes the basic ideas more transparent.
- (iii) **The infinitely repeated game can approximate finitely repeated games.** The results of infinitely repeated games often serve as a good approximation of the dynamics of finitely repeated games, especially if the number of rounds is above a critical threshold. The respective threshold can be very modest. In fact, as our additional simulations indicate, most of our key results already hold if players interact for ~10 rounds only. An infinite number of rounds is thus not required for our conclusions.

Our use of strictly alternating move structures can be explained analogously: (i) They allow for a more transparent comparison to simultaneous games. (ii) They lead to a simpler strategy space. (iii) They approximate the dynamics of more realistic games.

Having said that, we fully agree with the reviewer that it is important to quantify the impact of these different model assumptions. We provide such an analysis in our revised manuscript.

Changes: In the revised manuscript we treat the infinitely repeated game as our baseline scenario. In addition, we have extended both the main text and the SI to additionally discuss finitely repeated games and the impact of irregular alternation patterns. We describe the respective models in detail in our new **SI Section 3: Extensions of the baseline model**. We report the corresponding evolutionary results in our new **Figure 5, Figure S5, and Figure S6**.

Overall, the new results indicate that even for finitely repeated games, and even under irregular alternation patterns, the evolving strategies are qualitatively similar to the baseline model. Moreover, our results suggest that cooperation is most likely to evolve (i) when individuals interact for many rounds, and (ii) when they alternate in a fairly regular fashion. We thank the reviewer for pushing us to consider these model extensions in more detail.

Moreover, the authors allowed for only memory-1 strategies (remembering only the last decision of both, the player and the opponent) or reactive strategies based on only the opponent's last decision. Both, the setting and the restricted memory of players were the basis of great

work published in the early 1990s by Nowak and others. And obviously, a thorough game-theoretical analysis of cooperation strategies has to start with simplifying assumptions before complexity can be added step by step. But 25 years after these ground-breaking papers, analyses are limited if they are still based on these unrealistic assumptions.

Reply: In our original manuscript, we perhaps failed to clearly articulate what the key novelty of our paper is. The reviewer is correct that memory-1 strategies have been around for more than two decades. However, for most of this time, work was restricted to simulations. The respective studies highlighted various memory-1 strategies that succeed in different evolutionary competitions (e.g., Generous Tit-for-Tat, Win-Stay Lose-Shift). However, these studies were unable to characterize *all* memory-1 strategies that can sustain cooperation. In hindsight, we believe progress was limited because the field lacked proper mathematical tools. This only changed within the last ten years, starting with the influential work of Press & Dyson (PNAS 2012) and others (Ethan Akin, Stewart & Plotkin). In particular, Stewart and Plotkin (PNAS 2014) characterized all memory-1 Nash equilibria of the simultaneous game without errors. With our paper, we extend this analysis to the case of alternating interactions. The tools that we develop apply to both, games with and without errors. This kind of analysis has been an open problem for the last 25 years (in fact, for the simultaneous game with errors, the problem is still open).

We would also like to point out that although we focus on memory-1 strategies, the strategies that we find are Nash equilibria in the strongest possible sense. That is, the strategies that we identify are not only robust with respect to other memory-1 strategies. Instead, they are robust with respect to all possible strategies, even against strategies that remember an arbitrary number of past interactions.

Having said that, we fully agree that an evolutionary analysis of more complex strategies would be desirable. But even with the better methods and hardware that we have 25 years later, this task is not straightforward. To see why, let us note that although there are only 16 pure memory-1 strategies, there are 65,536 pure memory-2 strategies, and more than 10^{19} pure memory-3 strategies. Given that typical computer simulations today introduce $10^6 - 10^{10}$ mutant strategies during the course of a simulation, a systematic analysis of all memory-3 strategies seems infeasible for the next decade. However, a systematic analysis of all pure memory-2 strategies is within reach. This is what we do in our revised paper.

Changes: To address the reviewers' concerns, we have made two sets of changes.

First, we now explain in more detail what the key contribution of our paper is. In particular, this discussion makes it more clear why a characterization of all stable memory-1 strategies for alternating games has been an open problem for more than 25 years.

Second, we complement our evolutionary simulations for memory-1 strategies with additional analyses and simulations for pure memory-2 strategies. The respective framework is laid out in the new **SI Section 3: Extensions of the baseline model**. The corresponding results are illustrated in **Figure S7**. The key results are as follows:

- (i) Even among the pure memory-2 strategies, there is no evolutionarily stable strategy that can sustain full cooperation in the alternating game (although such strategies are available for the simultaneous game).

- (ii) Nevertheless, simulations suggest that cooperation can still evolve in the alternating game. However, the corresponding strategies are more “short-lived” compared to the simultaneous game (they are less robust with respect to invasions by mutant strategies).

Therefore, a title like “Robust cooperation in alternating interactions” is promising too much.

Reply and changes: We have revised the title. It now reads “Cooperation in alternating interactions with memory constraints”.

An important take-home message from earlier work (beginning in the early 1990s) was that probabilistic strategies are more powerful than deterministic ones. Indeed, it was immediately clear that the highly praised win-stay-loose-shift (WSLS) strategy has little relevance in people's social lives when played as a deterministic strategy. A player of deterministic WSLS would always cooperate (at a rate of 100%) after mutual defection. Humans don't. If they are forgiving and cooperative after mutual defection, it is only with a probability of $< 100\%$. There are other arguments in favor of the strength of probabilistic strategies, much of which has already been elaborated in earlier work. Thus, when Park et al.'s abstract ends with the emphasis that "... mutual cooperation ... required probabilistic strategies", the novelty value of the paper is clearly limited. We have known this for over two decades.

Reply and changes: We believe it is useful to distinguish between previous theoretical work and empirical observations.

We fully agree that empirically, there is not much evidence that humans adopt the strategy Win-Stay Lose-Shift (WSLS) in repeated prisoner's dilemma experiments (note that these experiments usually use the setup of simultaneous games). Thus, from the viewpoint of empirical work, it is indeed not surprising that humans use strategies that appear to be stochastic.

However, this does not diminish the theoretical importance of our work. Theoretical work is often concerned with the question of whether evolutionarily stable cooperation is feasible *in principle*. As shown by Lorberbaum (JTB 2002) and others (e.g., McAvoy and Nowak, PRSA 2019), any such evolutionarily stable strategy *must* be deterministic. The simultaneous game allows for such evolutionarily stable cooperative strategies (WSLS is one example). In contrast, we show that in the alternating game, no cooperative memory-1 strategy can be evolutionarily stable. We believe this is an important theoretical finding, irrespective of the empirical merits of WSLS in the simultaneous game.

If one assumes that the human sense of reciprocity is at least partly shaped by alternating interactions, our study may even help to explain why WSLS predicts human behavior rather poorly. After all, in alternating interactions, WSLS is not expected to evolve in the first place. Eventually, however, the relevance of WSLS for human behavior is an empirical matter. To avoid any speculation, we have rephrased the corresponding sentences accordingly. In particular, we do no longer stress that “mutual cooperation requires probabilistic strategies” (although the statement remains true). Instead, we highlight that alternating games do not allow for evolutionarily stable cooperation in the context of memory-1 strategies.

Once again, although the reviewer's comments are perhaps more critical than the comments of the other two reviewers, these criticisms are well taken. We believe they helped us to clarify our contribution, and to explore the robustness of our conclusions in more detail. We have addressed the reviewer's comments by extending both the main text and the SI, and by running many additional computer simulations (**Figure 5, Figures S5-S7**).

Reviewer #3.

This paper considers the evolution of cooperation through direct reciprocity with asynchronous interactions. Understanding the evolution of cooperation is a fundamental problem in evolutionary theory and most of the work that has studied this topic using direct reciprocity has focused on simultaneously interactions. Since many real-world interactions are not simultaneous it is important to consider direct reciprocity with alternating (or more general patterns of) interactions. The current paper therefore fills an important gap in the existing literature. This paper provides a carefully and convincing analysis of the evolution of cooperation through direct reciprocity with alternating interactions that significantly extends the previously known results. The paper is well written and thoughtfully organized, and contains a number of new and interesting results that contribute meaningfully to the field.

Reply: Thank you very much for this positive feedback!

I only have two minor comments.

1. The authors assume, as is common in the field, that there are infinitely many interaction rounds in the game (and that payoffs from different rounds are not discounted). Since one of the most obvious facts about real-world interactions is that they often consist of quite a small number of interactions (and/or have payoffs that are quite heavily discounted) it would be interesting to know if the authors have any results (even if only numerical) indicating how applicable the results of the current paper are to this situation.

Reply: We concur, our analysis of infinitely repeated games is concerned with a somewhat idealized scenario (a similar point was also raised by reviewer #2). We believe this analysis is useful, because infinitely repeated games often serve as a good approximation of relationships with many (albeit finitely many) interactions. Nevertheless, we certainly agree that further simulations on finitely repeated games would make for a valuable addition.

Changes: To address this suggestion, we have implemented additional simulations for games with a finite expected length. The respective framework is briefly explained in the main text, and discussed in full detail in the new **SI Section 3: Extensions of the baseline model**. In addition, the main results are illustrated in the new **Figure 5** and **Figure S5**.

In a nutshell, the key results of this additional analysis are as follows:

(i) Provided the number of rounds is sufficiently large, the results for finitely repeated games are largely comparable to our previous results for infinitely repeated games. In particular, we observe more cooperation in the simultaneous game (**Fig. 5a**). Moreover, the cooperative strategies that evolve in the simultaneous game are different from the strategies in the alternating game (**Fig. 5b**).

(ii) Infinitely repeated games approximate the qualitative dynamics of finitely repeated games already for a moderate number of rounds. In fact, the above observations already hold if players only interact for 10 rounds on average (or equivalently, if the game's continuation probability is 90%, see **Figs. 5c** and **Fig. S5**).

2. It is known for the simultaneous game that structured populations can significantly effect the evolution of cooperation (see for example, Brauchli et al., Evolution of Cooperation in Spatially Structured Populations, JTB 200, 1999, 405-417). It would be very interesting to know what the corresponding effects of population structure would be for the alternating case that the authors are considering in the present paper.

Reply: This is a great suggestion, and we are grateful to the reviewer for pointing us to the work of Brauchli et al. We agree that it is interesting to ask how their results extend to the alternating games that we study.

Changes: For our revised paper, we performed additional simulations following the basic framework of Brauchli et al. In particular, instead of well-mixed populations, we simulate the behavior of individuals who are arranged on a two-dimensional lattice. While a complete analysis of the resulting spatial dynamics provides material for an entire paper on its own, our new **Figure S5g-i** gives some qualitative insights (some further details are discussed in **SI Section 3: Extensions of the baseline model**).

Similar to Brauchli et al for the case of simultaneous games, we observe that population structure can enhance the evolution of cooperation in alternating games. For example, in **Fig. 5g**, we observe that alternating games can yield cooperation rates of ~80% for a relatively small benefit-to-cost ratio of $b/c = 2$. Interestingly, we also find that while simultaneous games tend to result in monomorphic populations, alternating games can sometimes lead to stable mixtures of cooperative and non-cooperative strategies (for snapshots of the population, see **Fig. 5h**). On the level of evolving strategies, we recover Brauchli's finding that simultaneous games further promote the evolution of Win-Stay Lose-Shift like strategies (**Fig. 5i**). For alternating games, evolving strategies show the main characteristics of Stochastic Firm-but-Fair and Generous Tit-for-Tat, as predicted by our previous results for well-mixed populations.

REVIEWER COMMENTS

Reviewer #1 (Remarks to the Author):

The authors have substantially revised the manuscript and incorporated my suggestions and those of the other reviewers. I am happy to support publication.

Reviewer #2 (Remarks to the Author):

The authors have well responded to all my points. I find the MS greatly improved and ready for publication. This paper will be of much interest to a wide readership.